behaviour, biological applications

parrots, animal welfare, captive breeding, stress, abnormal behaviour, phylogenetic comparative methods

**Author for correspondence:**
Georgia J. Mason
e-mail: gmason@uoguelph.ca

†These authors contributed equally to this study.
‡Present address: Lunenfeld-Tanenbaum Research Institute, Sinai Health, Toronto, Canada.

# Nature calls: intelligence and natural foraging style predict poor welfare in captive parrots

Emma L. Mellor[1,†], Heather K. McDonald Kinkaid[3,†,‡], Michael T. Mendl[1], Innes C. Cuthill[2], Yvonne R. A. van Zeeland[5] and Georgia J. Mason[4]

[1]Bristol Veterinary School, and [2]School of Biological Sciences, University of Bristol, Bristol, UK
[3]Formerly, Animal Biosciences Department, and [4]Department of Integrative Biology, University of Guelph, Canada
[5]Department of Companion Animal Clinical Science, Utrecht University, The Netherlands

(iD) MTM, 0000-0002-5302-1871; ICC, 0000-0002-5007-8856; GJM, 0000-0001-8045-4579

Understanding why some species thrive in captivity, while others struggle to adjust, can suggest new ways to improve animal care. Approximately half of all Psittaciformes, a highly threatened order, live in zoos, breeding centres and private homes. Here, some species are prone to behavioural and reproductive problems that raise conservation and ethical concerns. To identify risk factors, we analysed data on hatching rates in breeding centres (115 species, 10 255 pairs) and stereotypic behaviour (SB) in private homes (50 species, 1378 individuals), using phylogenetic comparative methods (PCMs). Small captive population sizes predicted low hatch rates, potentially due to genetic bottlenecks, inbreeding and low availability of compatible mates. Species naturally reliant on diets requiring substantial handling were most prone to feather-damaging behaviours (e.g. self-plucking), indicating inadequacies in the composition or presentation of feed (often highly processed). Parrot species with relatively large brains were most prone to oral and whole-body SB: the first empirical evidence that intelligence can confer poor captive welfare. Together, results suggest that more naturalistic diets would improve welfare, and that intelligent psittacines need increased cognitive stimulation. These findings should help improve captive parrot care and inspire further PCM research to understand species differences in responses to captivity.

## 1. Introduction

When kept by humans, why do some species thrive, yet others struggle? This question has been relevant since the dawn of domestication. Today, with wild populations under increasing threat, and captive populations dominating some taxa, it has both conservation and welfare implications. Many wild animal species enjoy impressive lifespans and breeding success when kept in zoos, breeding centres or people's homes [1]. Yet others are prone to behavioural, health and reproductive problems (e.g. [2,3]), raising ethical concerns when indicative of stress, and conservation concerns if captive populations become non-sustainable (e.g. [4–6]). Parrots illustrate such issues well. For this highly threatened order, in which > 40% of species are threatened or near threatened [7], captive populations equal wild ones in size (each around 50 million; J. Gilardi, World Parrot Trust, 2020, pers. comm.), spanning zoos (for a few thousand individuals [8]), breeding centres (for tens of thousands [9]) and private homes (for tens of millions [10]). Here, some species do well. Captive cockatiels, *Nymphicus hollandicus*, for instance, typically breed well [11] and show little evidence of stress [12,13]. Others, in contrast, despite living in similar conditions, are prone to disease (e.g. [14]), apparently shortened lifespans [15] and poor

reproduction (with yellow-faced Amazons, *Amazona xanthops*, among the many showing just a fraction of their wild reproductive outputs [8,16]). Such problems may reflect deleterious effects of genetic bottlenecks [17,18] and/or captivity-induced stress [18–20]. More specific evidence of stress comes from the abnormal, repetitive, 'stereotypic' behaviour [21–24] common in some species. For example, 10–15% of pet parrots show feather-damaging behaviour (FDB), chewing, plucking and/or ingesting their own feathers [12,25] in ways that compromise flying and thermoregulation, and even cause tissue damage [12,13,26]. FDB prevalence can be 40% in some species (e.g. grey parrots, *Psittacus erithacus*), yet in similar living conditions, other species (e.g. Senegal parrots, *Poicephalus senegalus*) rarely display it [12,13,27–29].

Phylogenetic comparative methods (PCMs) developed by evolutionary biologists can reveal why species show such variation. PCMs allow multi-species datasets to be statistically analysed to identify attributes that predispose certain taxa to problems in captivity, while controlling for similarities arising from evolutionary relatedness (e.g. [3,5]; see electronic supplementary material, figure S1 for an illustration). One potential such attribute is the degree of difference or 'mismatch' [30,31] between typical wild and captive environments. This determines the extent to which captivity constrains animals' natural behaviours, which can cause frustration (e.g. [32,33]). A second is phenotypic plasticity, which, if high, can enable animals to adjust to such mismatches (e.g. [2,5,31]). A third is a species's conservation status (endangeredness), which can predict vulnerability to both poor welfare [2,31] and problems induced by small population sizes *per se* (e.g. inbreeding [34]). All three potential risk factors make predictions that are testable in parrots.

For parrots, major mismatches exist between captive and wild environments in social complexity, and in the opportunities available for foraging, decision-making and cognitive problem-solving. Most parrots are naturally highly social [35,36], and social contact is often crucial for animal welfare [37,38], yet captive parrots frequently live with little or no access to conspecifics [35,39,40]. If this causes stress, then naturally highly social species should be most at risk of welfare problems in captivity. Turning to foraging, wild parrots spend 40–75% of their active time in this behaviour [41–43], yet captive birds face spatial restrictions, dish feeding and processed diets which constrain opportunities to search for, select and manipulate food [40,44], a mismatch suggested to reduce parrot welfare (e.g. resulting in FDB [13,26,45]). This hypothesis predicts that species with naturally time-consuming foraging will be most at risk of welfare problems. A final major mismatch is that captivity constrains opportunities to explore, make decisions and problem-solve (e.g. [31,37,46]). This has been argued to compromise welfare in large-brained, intelligent species (e.g. great apes, elephants and cetaceans [31,47,48]), and in naturally generalist species said to risk 'boredom' in captivity [49,50]. Parrots often resemble primates in relative brain size (i.e. encephalization), neuron number and cognitive abilities [51,52]. Many are also naturally opportunistic omnivores. This third mismatch hypothesis thus predicts that captive species with high degrees of generalism and intelligence will be most at risk of welfare problems.

However, intelligence and generalism could instead be *protective*, because they promote behavioural plasticity. In the wild, large-brained innovators are good at coping with human-induced rapid environmental change, HIREC (e.g. urbanization, translocation and climate change), thanks to their behavioural flexibility [31,53–56]. Further forms of plasticity are protective too: species with broad dietary or habitat niches have proved easier to domesticate, and cope better with HIREC than those with narrow niches [57–59]. Consistent with this idea, one author proposed that 'less specialised animals … settle down most easily in zoos, and exhibit less disturbed behaviour' [60]. The hypothesis that behavioural plasticity pre-adapts animals to adjust to captivity [31] thus generates an alternate set of predictions about traits conferring risk, with small-brained, specialist parrots being most prone to welfare problems.

The final type of potential risk factor is an endangered conservation status. Traits common in threatened species—especially timidity, wide-ranging lifestyles and low behavioural plasticity—may put them in 'double jeopardy': both vulnerable in the wild *and* at risk of stress when captive [2,31]. Furthermore, the small population sizes of rare species can cause further issues, such as the increased expression of deleterious alleles (e.g. [17,61]). Consistent with such potential problems, endangered bird species may be harder to breed than their non-endangered relatives [62], and endangered parrot species are also disproportionately rare in private breeding centres [9], zoos [63] and the pet trade [7]. We therefore tested two further hypotheses: that rarity predicts breeding problems, and that threatened parrots are most at risk of captivity stress.

## 2. Methods

### (a) Outcome variables: stereotypic behaviour and captive hatch rates

To test these hypotheses, we sought high-quality data for multiple species (required when using PCMs). For stereotypic behaviour (SB), we looked to the largest captive sector, pet parrots [8–10], running an online survey for pet owners for 15 months (www.parrotsurvey.com). Following quality controls (see below and electronic supplementary material), this generated data on 1378 individuals across 50 species. For each species, we calculated the prevalence of FDB, other oral SBs (e.g. biting/mouthing the cage bars) and SBs involving the head or body (e.g. route-tracing, head-twirling). The survey also collected detailed information on species-typical demographic and husbandry characteristics (see below and electronic supplementary material, table S1), which could potentially be highly influential and even act as confounds (see 'Statistical procedures and analyses')

For reproductive success, pet owners could not supply data as they seldom bred their birds. Instead, we obtained captive hatch rates from the Psittacine Captive Breeding Survey, a 1991 census of over 31 000 parrots in 1183 private breeding collections by TRAFFIC USA [9]. Their contemporary relevance was confirmed via independent ratings from a present-day aviculturalist (see electronic supplementary material). These records enabled us to calculate chicks hatched per breeding pair per year for 122 species, which, after quality controls (see electronic supplementary material), yielded values for 115 species. To control for species differences in life history [5,58,64], data on natural fecundity were obtained for inclusion in models (product of the median eggs per clutch and clutches per year [16,65]; see 'Statistical procedures and analyses'). All species-typical values for SB pet population characteristics and breeding output in private breeding centres are given in the electronic supplementary material, table S2.

**Table 1.** Details regarding potential predictors of behavioural and reproductive problems in captive Psittacines: data sources, how quantified and the effects predicted by each hypothesis under test.

| hypothesis: behavioural and reproductive problems in captivity reflect... | predictor variable(s) | predicted relationship with behavioural and reproductive problems if hypothesis is correct |
|---|---|---|
| ...constraints on natural social interaction | Data on social bond strength/number were unavailable, and so group size was used as a proxy (cf. [38]). *Maximum feeding group size*: maximum feeding group size during the non-breeding season, a measure of sociality while active [36,66]. | positive |
|  | *Communal roosting*[a]: whether or not a species roosts communally while sleeping [36]. | yes > no |
| ...restriction of natural foraging behaviours | Frustrating appetitive and consummatory aspects of foraging can give rise to stereotypic behaviour [67,68]. Quantitative wild time budget data were unavailable, so time spent on food-search and/or -handling [cf. 69] in the wild was inferred from diet-types reported in the EltonTraits avian foraging database [70] and other sources [65]. |  |
|  | *% natural diet requiring extensive locomotor/visual food search*: reliance on food items that are naturally patchily distributed in space and time, and/or scarce or inconspicuous (based on [71–73] and adapted here in line with EltonTraits' dietary categories), namely invertebrates, nectar and pollen, fruit, and tree seeds/nuts. | positive |
|  | *% natural diet requiring extensive food handling*: reliance on food items requiring extensive oral manipulation (based on [71–73] and adapted here), namely invertebrates and tree seeds/nuts. (Note that Wilman *et al.* [70] pooled reliance on small/grass seeds and tree seeds/nuts into one category, "seeds", but we made opposing predictions about each regarding welfare. Therefore, by referring to their literature sources [65], and following their methods, we split each species' reliance on seeds/nuts proportionally between small/grass seeds and tree seeds/nuts[b]). | positive |
| ...being a habitat or dietary generalist | *Diet breadth*: total number (1–5) of main food types in the species-typical native adult diet [16,65], from: seeds/nuts, fruits/berries, pollen/nectar, other vegetative material, and animal material [58,59]. | Negative or positive depending on whether generalism is protective or a risk factor |
|  | *Habitat breadth*: total number (1–7) of main habitat types in the native range, from: mixed lowland forest, alpine scrub and forest, grassland and savannah, mixed scrub, marsh and wetland, cultivated and farmland, and urban environments [59]. |  |
| ...being intelligent and innovative | *Innovation frequency*: total number of feeding innovations reported by regions [74–77], supplemented with unpublished data (Louis Lefebvre, pers. comm., 2013)[c]. *Relative brain volume [encephalization]* was used as a proxy for intelligence ([78–81]; see additional information in Discussion). A species' average endocranial volume (ml) taken from skeletal specimens or converted from brain mass [82,83][d]. | Negative or positive depending on whether intelligence is protective or a risk factor |
| ...being rare/threatened | *IUCN Red List category*: status in the International Union for Conservation of Nature Red List of Threatened Species [87] (ranked 1–5): Least Concern, Near Threatened, Vulnerable, Endangered, Critically Endangered. | Positive |
|  | *Population sizes* in US aviculture systems: number of pairs per species, taken from [9]. | Positive (*N.B. only reproductive problems assessed here*) |

[a]The one categorical (rather than continuous) predictor.

[b]Two species' diets were recorded incorrectly in EltonTraits: the black-headed parrot *Pionites melanocephalus*, was coded as using 60% nectar, yet its source account [65] did not mention it using nectar but rather tree seeds; and the dusky parrot, *Pionus fuscus*, was coded as 100% fruit but according to the source [65] it uses seeds from two trees. On advice from EltonTraits' authors (Y. Belmaker, pers. comm., 2020), we corrected these entries for our dataset.

[c]Research effort (number of published papers on a given species) was included in all models to control for differential interest by birdwatchers [76], calculated from results of searches of species names (scientific and common) in the 'Topic' field of the Zoological Records web index (Thomson Reuters) between 1978–2004 [76,77].

[d]To control for allometry [83–85], each species' average body mass (g) [82,83] was included in all models. We also excluded values taken from single animals [cf. 86].

## (b) Potential predictors: captive-wild mismatches, traits conferring behavioural plasticity and endangeredness

We collated data on sociality in the wild (maximum group size when foraging; use of communal roosting when sleeping); reliance on extensive foraging in the wild (inferred from the percentage of the natural diet requiring prolonged food search or handling); and indicators of behavioural plasticity (habitat and diet niche breadths; feeding innovation rate; relative brain size [encephalization], a marker of intelligence). Our two measures of endangeredness were threat level according to IUCN Red List categories and *ex situ* rarity: captive population sizes in private breeding centres. Descriptions of each are provided in table 1 (plus sources used); species-typical values are given in the electronic supplementary material, table S3 (with fuller details available via links provided in the electronic supplementary material).

## (c) Statistical procedures and analyses
### (i) Data analysis: general methods

All analyses were performed in R [88], with alpha set at 0.05. To control for species non-independence (e.g. [89–91]), we used phylogenetic generalized least-squared (PGLS) regressions in 'caper' [92] for continuous outcomes and phylogenetic logistic regressions in 'phylolm' [93] for binomial outcomes. Before hypothesis-testing, we constructed a consensus phylogenetic tree (electronic supplementary material, figure S2) from 1000 BirdLife parrot trees (Hackett backbone) [94,95] in 'phytools' [96]. We first used this to run PGLS models for our 50 pet species assessing potential confounds between predictor variables and key captive population characteristics that could influence SB (e.g. lack of enrichment, being housed alone; see electronic supplementary material, table S1). Along with a treeblock, it was then used in hypothesis-testing models as described next.

### (ii) Hypothesis testing

Each predictor was regressed against our outcome variables (table 1 for details). Following [97], any captive population characteristic emerging as significantly confounded with a predictor variable (see electronic supplementary material, table S4) was included in the relevant hypothesis-testing model for SB (see below). In all breeding centre hatch rate models, natural fecundity was included to control for species differences in life-history traits [5].

To check that any findings were robust, we then added three further steps:

(i)  We investigated relationships between the predictor variables used to test different hypotheses (see electronic supplementary material, table S5), then assessed whether any collinearity detected could artefactually account for or obscure any initial findings.
(ii) To account for phylogenetic uncertainty [98], analyses were repeated over a 'tree block' of the 1000 parrot phylogenetic trees. This improved parameter estimation and generated 95% confidence intervals (CIs) (e.g. [94,98]).
(iii) To assess whether any significant findings merely reflected outliers, we used a custom version of the influ_phylm function within the 'sensiPhy' package [99]. This performs 'leave-one-out' deletion analyses (removing each species in turn and recalculating the intercept, slope and corresponding *p*-value for each parameter), a species being deemed 'influential' if its removal yielded a standardized difference > 2. This revealed whether any results critically relied on the influence of just one or two particular data points.

## 3. Results

Different species varied greatly in the prevalence of SB and how their captive hatch rates compared to wild fecundity (see electronic supplementary material, table S2); some hypothesized predictors explained this variance. Summarized results of all final hypothesis-testing models are shown in table 2 (with full model outputs presented in electronic supplementary material, table S6, and further details in electronic supplementary material, table S7). Tests of our 'mismatch' hypotheses yielded the following. Constraints on natural social interaction did not seem to predict problems in captivity: naturally more social species were not more at risk of more SB or poorer captive hatch rates. Restricting natural foraging behaviours, in contrast, emerged as important for abnormal behaviour: species naturally reliant on diets requiring extensive handling had a higher FDB prevalence (partial $R^2 = 0.16$; table 2 and figure 1), a result that 'leave-one-out' analyses showed was robust (electronic supplementary material, table S7). Furthermore, there was evidence that one type of behavioural plasticity is an additional risk factor: species with relatively large brains had higher prevalences of both whole-body and oral SB (partial $R^2 = 0.19$ and 0.26 respectively; figure 2 and table 2). 'Leave-one-out' analyses showed that both these effects were also extremely robust (see electronic supplementary material, table S7).

Finally, turning to endangeredness (rarity and threat level), these did not significantly predict SB (table 2; electronic supplementary material, table S6). However, the number of breeding pairs present within each species's aviculture population did predict captive hatch rates, small populations having significantly poorer reproductive outputs (table 2; electronic supplementary material, table S6). This result was extremely robust to species' removal (electronic supplementary material, table S7). IUCN Red List category also appeared to predict captive breeding success, with more endangered species having lower hatch rates, if the confound of foraging group size was statistically controlled for (table 2; partial $R^2 = 0.06$). However, this result was vulnerable to species' removal, reliant on the influence of just two key species; it was thus not robust.

## 4. Discussion

Our results confirmed that PCMs can test otherwise intractable hypotheses about causes of poor welfare, so yielding new insights for improving wild animal care and captive breeding. The strongest effect in our data was that psittacines with relatively large brains were most prone to two categories of SB. Encephalization, our marker of intelligence, thus explained over 25% of the variance in oral SBs not directed at feathers (e.g. repetitive biting at cage bars), and nearly 20% of the variance in whole-body forms like route-tracing (the most common SB, affecting 45 of our 50 pet species and 23.2% of individuals). A larger relative brain size reflects a larger pallium, which is associated with general cognition [84] (and homologous with the neocortex, critical for general intelligence in primates [81]). It predicts greater behavioural flexibility in the wild (e.g. [53]) and thence improved establishment success, even invasiveness, in bird populations translocated to novel wild environments [53,54]. Being placed in a novel *captive* environment clearly poses a very different challenge for

**Table 2.** Species-level predictors of behavioural and reproductive problems in captive parrots. Summary of key results from hypothesis-testing PCM models. Each partial $R^2$ shown is calculated from the median values for the term's t-statistic (over the tree block); its associated degrees of freedom is also reported, with + and − denoting the direction of the relationship (in brackets if non-significant). Significant ($p < 0.05$) results are shown in **bold**. Full model results are shown in electronic supplementary material, table S6. ED = enrichment diversity; prop. = proportion.

| Hypothesis: behavioural and reproductive problems in captivity reflect... | predictor: | SB results — captive population characteristic(s) controlled for[c] | whole-body SB | FDB | oral SB | hatch rate results[d] |
|---|---|---|---|---|---|---|
| ... constraints on natural social interaction | maximum feeding group size | | (+) $R^2 = 0.03$ | − $R^2 = 0.10$ | (−) $R^2 < 0.01$ | (+) $R^2 = 0.02$[e] |
| | communal roosting (yes versus no) | current ED; early ED | (−) $R^2 < 0.01$ | (+) $R^2 < 0.01$ | (+) $R^2 < 0.01$ | (−) $R^2 = 0.01$ |
| ... restriction of natural foraging behaviours | % natural diet needing extensive search | prop. standard cage; captive diet diversity | (+) $R^2 = 0.02$ | (+) $R^2 = 0.01$ | (+) $R^2 < 0.01$ | (−) $R^2 = 0.03$[f] |
| | % natural diet needing extensive handling | prop. adult; prop. female; prop. standard cage | (+) $R^2 = 0.01$ | **+ $R^2 = 0.16$** | (+) $R^2 = 0.06$ | (−) $R^2 = 0.01$ |
| ... habitat and dietary generalism | diet breadth (1–5, count of main food types) | | (+) $R^2 = 0.04$ | (−) $R^2 < 0.01$ | (+) $R^2 = 0.02$ | (+) $R^2 = 0.01$ |
| | habitat breadth (1–7, count of main habitat types) | | (+) $R^2 = 0.04$ | (+) $R^2 = 0.10$[g] | (−) $R^2 < 0.01$ | (+) $R^2 = 0.02$ |
| ... being intelligent or innovative | innovation rate[b] (number reported) | prop. hand-reared | (−) $R^2 = 0.02$ | (+) $R^2 = 0.02$ | (+) $R^2 < 0.01$ | (−) $R^2 = 0.01$ |
| | brain volume[a] (ml) | prop. standard cage | **+ $R^2 = 0.19$** | (+) $R^2 = 0.02$ | **+ $R^2 = 0.26$** | (−) $R^2 < 0.01$ |
| ... being rare/threatened | IUCN Red List category (1–5, where 1 is Least Concern and 5 is Critically Endangered) | | (+) $R^2 < 0.01$ | (+) $R^2 = 0.01$ | (+) $R^2 = 0.01$ | (−) $R^2 = 0.03$ (see main text for additional analyses) |
| | no. of pairs in aviculture dataset | | N/A | N/A | N/A | **+ $R^2 = 0.08$** |

The following controls were included (see text and electronic supplementary material for more details):

[a]Body mass included in all models to control for allometry;

[b]Research effort included in all models to control for this potential influence on observed innovation rates;

[c]Species-typical attributes of pet populations controlled for where these covaried with a predictor variable;

[d]Natural fecundity included in all models to control for life-history traits.

In some models, the following correlated predictors were also included because collinear with the predictor under investigation (see electronic supplementary material, table S5):

[e]IUCN Red List category.

[f]Brain volume.

[g]Maximum feeding group size.

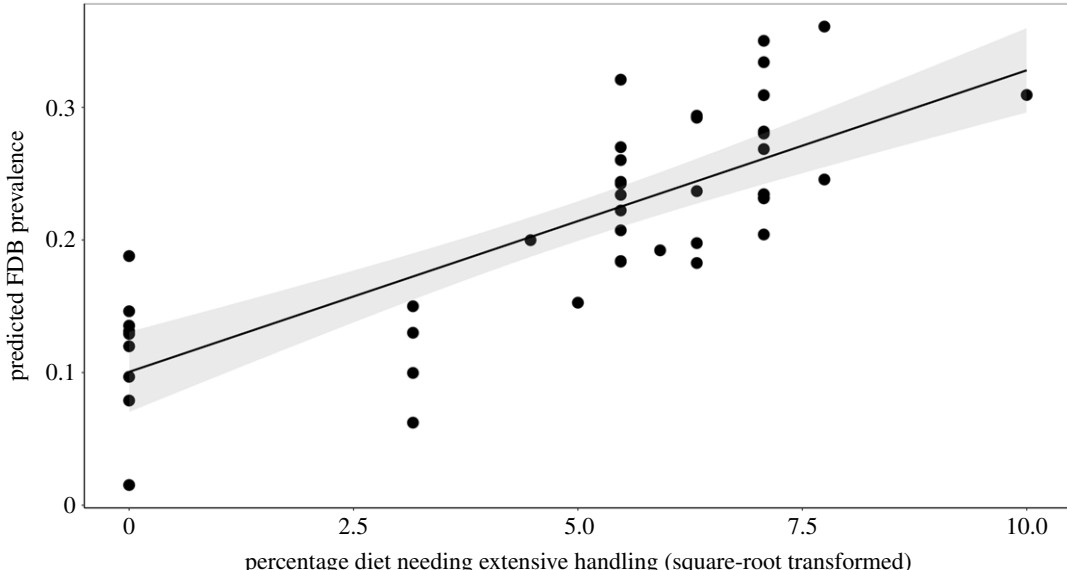

**Figure 1.** Species reliant on wild diets needing extensive handling have more prevalent FDB. Each data point is a species; the model contained other predictor terms (table 2; electronic supplementary material, table S6), and so predicted rather than raw values of FDB are shown; the shaded area shows the 95% CIs of the slope.

parrots: one where intelligence is a harm rather than a benefit. Species particularly prone to these SBs thus included monk parakeets, *Myiopsitta monachus*, highly successful invaders in the wild [100–102], and blue-and-yellow macaws, *Ara ararauna*, whose forebrains are more neuron-rich than those of rhesus monkeys, *Macaca mulatta* [51].

Despite strongly predicting SB in birds kept as pets, encephalization did not predict poor reproduction in breeding centres. We suspect this reflects the differential sensitivity and specificity of these two welfare indicators. SBs have very good specificity, reliably indicating sub-optimal rearing (e.g. [103]), aversive current treatments [85,104] and recurrent stressors over the lifespan (e.g. [85]). However, SBs have poor sensitivity: their absence does not always indicate good welfare, because inactivity is an alternative response in some genotypes (e.g. [105,106]). Our results could thus reflect that larger- and smaller-brained species do not differ in their *degrees* of stress, but instead in how they display it, with smaller brained species expressing poor welfare hypoactively (e.g. with apathy) instead of hyper-actively via SB. Alternatively, however, that relatively large brains did not predict poor reproduction could well be a false negative result. This seems likely because, while reproductive problems can reflect stress [20,107], they have poor specificity as stress indicators, being affected by multiple other factors (including genetic bottlenecks [17,61], a lack of appropriate imprinting opportunities and artificial incubation (e.g. [117]), all factors that would add noise to our analyses which we could not parse out (so increasing chances of Type II error). Furthermore, recent evidence suggests that parrot species which commonly breed in captivity tend not to be invasive if released [102], consistent with them being smaller brained, and that birds become less encephalized with domestication (e.g. [108]), consistent with large brains being selected against in captivity. We therefore suspect that the lower SB of smaller-brained species *does* indicate that they adjust better to captivity than do larger-brained species.

Further work is now required, but these findings still provide the first empirical evidence that intelligent species can have unmet welfare needs in captivity. That captive conditions can be predictable, monotonous and unchallenging is often argued to reduce well-being in intelligent mammals (e.g. [31,47,48]), and intelligent species often attract special welfare protection e.g. those afforded to primates over other mammals (e.g. [109] when used in research). Our results suggest such concerns are well-placed and should now extend to large-brained birds, with parrots (perhaps also corvids [110]) being given care better tailored for intelligent species. As for precisely *what* large-brained species lack in captivity, opportunities for exploration, learning and agency are all possibilities. Their absence could cause the diverse array of SBs shown by large-brained parrots by compromising normal brain development [21], by promoting boredom (aversive states caused by monotony [37,111]) and resulting attempts to self-stimulate [104,106]), and/or by enhancing birds' motivations to escape (as underlie some mammalian SBs [21]; a concerning possibility given these larger-brained parrots' invasiveness). Opportunities for cognitive stimulation must therefore urgently be investigated, to identify the most effective ways to improve these parrots' well-being (e.g. [46,112]).

FDB, which affected 20.8% of our population, was instead predicted by naturally relying on food items in the wild that need extensive handling (inferred from diet type, since cross-species data on handling time budgets were unavailable). This metric explained 16% of the variance in FDB prevalence. One likely explanation is that parrots remain motivated to perform food handling even when captive diets do not require it, then redirecting these movements to their own feathers [21,26,113]. Alternatively, captive diets may lack the nutrients or cellulose/chitin present in nuts, tree seeds and invertebrates (potentially then altering the gut microbiome), parrots then ingesting feathers in attempts to rectify these deficits (cf. such effects in hens, *Gallus gallus domesticus*: [114,115]). More research is now needed to identify the precise mechanisms underlying FDB, but for now this finding highlights the likely importance of less processed, more naturalistic diets for parrot welfare; helps explain why foraging enrichment seems one of the more effective strategies for tackling parrot FDB [116]; and echoes experimental

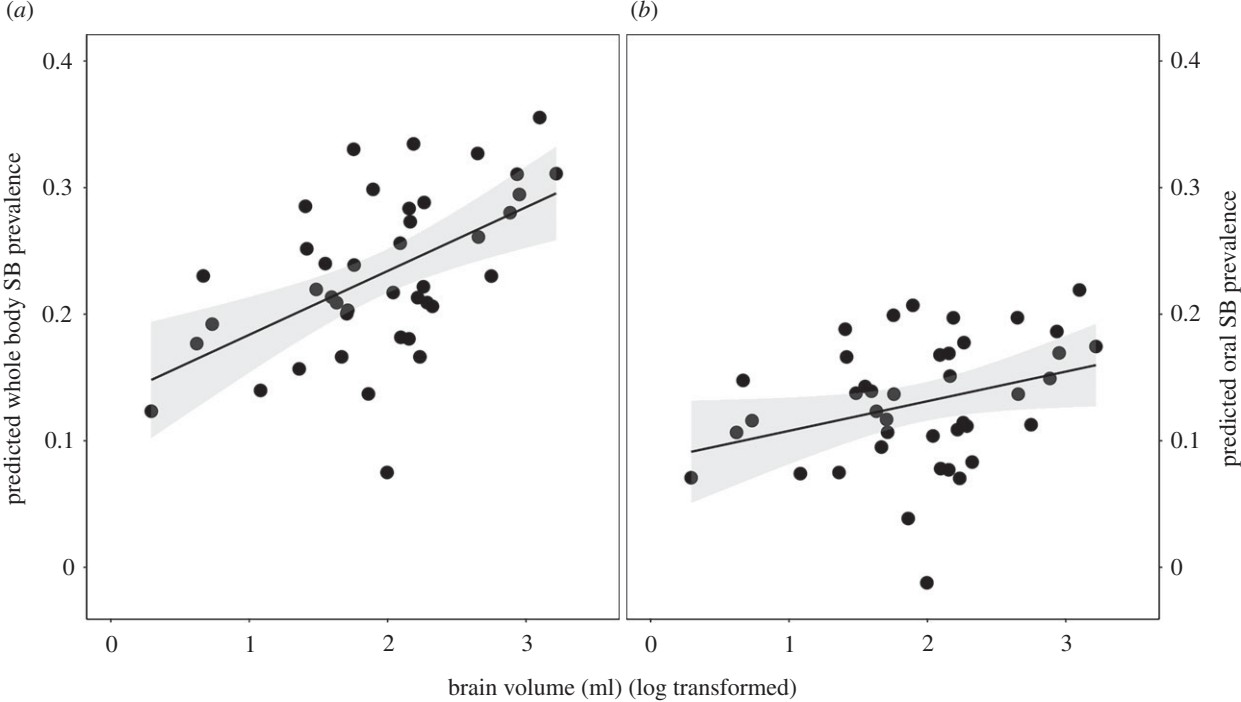

**Figure 2.** (*a*) Species with large brains (relative to body mass) have more prevalent whole-body SB. (*b*) Relatively large-brained species also have more prevalent oral SB. See figure 1 legend for explanation.

work in hens showing that feather-pecking involves the fixed action patterns used in feeding [113].

Turning to reproduction, we found no evidence that breeding success is affected by the degree to which captivity restricts natural behaviours. Further research is needed to determine whether this is a true negative, or (as we suspect) a false one caused by the multiplicity of other influences on captive reproduction. Nevertheless, we did find cautious support for an untested hypothesis raised three decades ago. Derrickson & Snyder [62] suggested that endangered species are harder to breed in captivity than their non-endangered relatives (their examples including whooping cranes, *Grus americana*, being harder to breed than their common relatives, greater sandhill cranes, *Antigone canadensis tabida*). Their hypothesis was plausible in part because small captive populations are so vulnerable to genetic drift and inbreeding [17,18,61], and to shortages of compatible mates and other management issues [117]. Consistent with this, our one robust finding related to population size: species represented by only a few pairs in captivity had the poorest reproductive rates relative to natural fecundity. How and why rarity and poor captive breeding are linked requires further study—ideally in present-day breeding centres, and incorporating data on the various management practices that could obscure underlying stress effects.

In sum, we successfully used PCMs to test hypotheses about the risks to captive psittacines posed by key discrepancies between natural and captive environments, degrees of behavioural plasticity, and aspects of rarity. As a highly threatened order, in which half of all individuals live in captivity, ensuring good captive welfare is a conservation imperative. In private homes (a sink of tens of millions of non-breeding individuals), a large proportion of birds show SBs indicative of poor welfare. To prevent these, our results suggest that care should improve to supply these wild animals with naturalistic food items and cognitive stimulation. If redressing these deficits is impossible, then perhaps the keeping of (potentially invasive) intelligent species with naturally

handling-intensive diets should cease. Further, the significant impact of encephalization indicates for the first time that in this taxon, intelligence, which is so protective in the wild, is the opposite in captivity. Whether similar effects operate across captive primates and cetaceans should now be investigated: topics ideal for future PCMs. Further work is also needed to understand why small captive population sizes predict low hatching rates, and what else explains the substantial variation in captive reproductive success across parrots. This is highly urgent because, even today, avian captive breeding centres often 'suffer notably high levels of hatching failure' [117]. More broadly, we recommend that PCMs—with their abilities to interrogate multi-species datasets and address hypotheses that would otherwise be challenging to test—are increasingly used to understand why it is that some species thrive, yet others struggle, when kept in human care.

**Ethics.** All aspects of this study were approved by the University of Guelph Research Ethics Board (REB number 11JL024).

**Data accessibility.** Data in Excel, and R codes in fill, are available from the Dryad Digital Repository: https://doi.org/10.5061/dryad.ns1rn8psb [118].

The data are provided in the electronic supplementary material [119].

**Authors' contributions.** E.L.M.: data curation, formal analysis, investigation, methodology and writing-original draft; H.K.M.K.: data curation, formal analysis, investigation, methodology, writing-review and editing; M.T.M.: funding acquisition, supervision, writing-review and editing; I.C.C.: methodology, supervision, writing-review and editing; Y.R.A.v.Z.: data curation, methodology, resources, writing-review and editing; G.J.M.: conceptualization, funding acquisition, investigation, methodology, project administration, supervision, writing-original draft, writing-review and editing

All authors gave final approval for publication and agreed to be held accountable for the work performed therein.

**Competing interests.** We declare we have no competing interests

**Funding.** NSERC is gratefully acknowledged for funding to G.J.M. and H.K.M.K. E.L.M. was funded by a University of Bristol PhD Scholarship and UFAW.

**Acknowledgements.** Many thanks to Nico Schoemaker for help with survey design; the survey translators who helped us reach dozens of countries; the hundreds of survey respondents; and Michael Kinkaid for extracting and processing the raw survey data. Many thanks to Rosemary Low and Susan Clubb for discussions about parrot aviculture and advice on our captive reproduction data. Many thanks to Claudia Mettke-Hofmann and Cathy Toft for advice on natural behavioural biology, especially parrots' natural foraging behaviour; Louis Lefebvre for data on innovation rates; Tim Wright for early advice about phylogenetic trees; and Jamie Gilardi for conservation perspectives. Many thanks to Miquel Vall-Llosera for supplying example R code for tree block analyses, and Gustavo Paterno for the custom version of influ_phylm for the 'leave-one-out' analyses. Thanks to H.M.K.'s advisory committee: Dave Barney, for his enthusiasm and captive breeding wisdom, and Lee Niel, for her encouragement and survey expertise. Also thanks to two thoughtful referees for their comments. The University of Guelph authors acknowledge the ancestral lands of the Attawandaron people and the treaty lands and territory of the Mississaugas of the Credit.

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
