## [Peer Review File · Proceedings of the Royal Society B: Biological Sciences]

Review History

Decision letter (RSPB-2021-0899.R0)

28-Apr-2021

Dear Professor Mason:

Thank you for submitting your manuscript RSPB-2021-0899 entitled "Nature calls: Intelligence and natural foraging style predict indicators of poor welfare in captive parrots" to Proceedings B.

All manuscripts are assessed by a specialist member of the Editorial Board, who decides whether the manuscript is suitable for Proceedings B.

Unfortunately, your manuscript has been rejected at this stage of the assessment process. Competition for space is currently extremely severe, and we receive many more manuscripts than we are able to publish. On this occasion it was felt that your manuscript was unlikely to be able to compete successfully for a space in the journal.

Please find below the specialist Board member's comments. I hope you may find these useful should you wish to submit your manuscript elsewhere.

Sincerely,
The Proceedings B Team

Board Member

Comments to Author(s):

This study used Phylogenetic Comparative Methods (PCMs) to investigate predictors of abnormal behaviour and poor reproduction in captive parrots in relation to the mismatch between captive and wild environments, finding that those species with larger relative brain sizes (a proxy for intelligence) were most vulnerable to exhibiting stereotypic behaviour, and that feather damaging behaviour appeared linked to inadequate captive diets. The relationship between the cognitive capabilities of any given animal species and its welfare requirements is an interesting one, but the methodological approach taken and the results presented here needed to be more convincing. For instance, the reliance on owner-derived data, together with significant proxy-data inferences (e.g. handling of food items) and aged (reproductive success) data sets weakened the conclusions that could be drawn.

RSPB-2021-1027.R0

Review form: Reviewer 1 (Paul Rose)

Recommendation

Accept with minor revision (please list in comments)

Scientific importance: Is the manuscript an original and important contribution to its field?

Excellent

General interest: Is the paper of sufficient general interest?

Acceptable

Quality of the paper: Is the overall quality of the paper suitable?

Good

Is the length of the paper justified?

Yes

Should the paper be seen by a specialist statistical reviewer?

No

Do you have any concerns about statistical analyses in this paper? If so, please specify them explicitly in your report.

No

It is a condition of publication that authors make their supporting data, code and materials available - either as supplementary material or hosted in an external repository. Please rate, if applicable, the supporting data on the following criteria.

Is it accessible?

Yes

Is it clear?

Yes

Is it adequate?

Yes

Do you have any ethical concerns with this paper?

No

Comments to the Author

A useful and relevant paper that provides some clear applications to industry and the keeping of animals based on ecological knowledge. This is a well-written and well structured paper that has novel and interesting findings. It would benefit from some clarity in the explanations of terms used, the details provided in the statistical modelling and more consideration given to the application of results.

A couple of minor points for consideration:

Take with statements that "this is the first to show something" as there may be evidence somewhere that could come to light in the future.

Avoid the word prove. We support or refute in scientific research.

I can see that you like the term "thrive or fail" but to me, fail does not seem the correct wording for such a statement. Species can fail to adapt to a captive population, but in this sense it is not the individual's failure to cope with its captive husbandry. The husbandry is failing the individual. The animal is not at fault. Please change this word (wherever it occurs in the manuscript).

The abstract provides some really useful explanation of the key results but the concluding statement is quite weak. Could you suggest how they are applied to parrot care?

Also, suggest you remember the word "novel" (findings) because they are currently novel but the paper will age.

The introduction is clear and well explained. Is the word "dogged" correct (line 4)? Something more scientific and less colloquial sounds more appropriate.

The size of the captive parrot population seems huge. How measured? Is this including the budgerigar? And is this an estimation for the entire order?

Species comparison for the grey parrot could be helpful (line 22).

I appreciate that Phylogenetic Comparative Methods are a specialism of this research group but could the authors provide a bit of definition on what they are and what they do? Perhaps something along the lines of the useful diagram (Figure 1) in Mellor et al. 2018 Zoo Biology but using parrots as example species?

Line 30: I read endangeredness as a threat level but this relates to the likelihood of a poor welfare problem being experienced? But then I get to line 61 and see it is related to conservation threat? Please can this be explained more carefully?

If you are discussing how threatened a species is, threat level is probably a better term than "endangeredness".

Line 47: Perhaps explain boredom?

Methods are clear and repeatable and the supplementary information is useful for their understanding.

Line 97: Can you explain the key source for social behaviour?

Line 101: Please explain or define what you mean by encephalisation in this section. Is brain size a proxy for this?

Line 103: Where do captive population sizes come from?

Is aviculture defined as captive populations in managed programmes, i.e. those run by accredited or member organisation zoos? Or for private breeders too?

Line 132: I am not familiar with "leave-one-out" analysis. Can this be explained?

For all models, can you publish the final model that was run?

Line 153: I am struggling to work out why you decide to include and leave out species. What's the biological or ecological decision behind this, and the long term impact on your results?

Line 156: weak effects on...?

Line 157: suggest you explain the correlated predictor here, how strong this was, and what you did.

Figure 1 is useful. How is extensive handling time defined?

Line 189 suggest that you don't lead this paragraph with the ambiguous result. Present the clear result first, that small populations and number of breeding pairs clearly predicted hatch rates. Threat level is unclear. I don't think it is appropriate to try and draw a conclusion from the threat level data given your own explanation of how sensitive it is to specific data points.

Line 160: because the grey parrot is so common in captive populations, is this why it has such an influence? So is the small population of each individual species in this large but varied dataset causing issues with the reliability of the analysis output?

The discussion is thorough and provides a detailed evaluation of results. I would like to see more application of the findings to the actual birds that will benefit from them. Currently this is glossed over or left to the reader to infer.

Line 226-227: I find this sentence hard to follow. And please explain your reasoning for the Type II error.

Line 240-241: Please explain what you mean by better care, and provide some examples of how this can be done (based on your results). Less sunflower seeds, more environmental complexity, for example.

Line 254: Is there a reference for this? Parrots on complete diets can still show abnormal behaviours. How relevant is the comparison to the domestic hen?

Line 272: Perhaps because of genetic drift over time? You compare this finding to the published literature in the crane example stated previously, but I would state your findings and then do the comparison.

Line 291: Why should PCM be used more? Could you add context to the end of your conclusion?

Review form: Reviewer 2

Recommendation

Major revision is needed (please make suggestions in comments)

Scientific importance: Is the manuscript an original and important contribution to its field?

Excellent

General interest: Is the paper of sufficient general interest?

Excellent

Quality of the paper: Is the overall quality of the paper suitable?

Good

Is the length of the paper justified?

Yes

Should the paper be seen by a specialist statistical reviewer?

No

Do you have any concerns about statistical analyses in this paper? If so, please specify them explicitly in your report.

Yes

It is a condition of publication that authors make their supporting data, code and materials available - either as supplementary material or hosted in an external repository. Please rate, if applicable, the supporting data on the following criteria.

Is it accessible?

Yes

Is it clear?

Yes

Is it adequate?

Yes

Do you have any ethical concerns with this paper?

No

Comments to the Author

General:

Throughout the manuscript the term captive is utilized when referring to many different situations. This includes pets, aviculture, zoos, etc. and the manuscript could benefit from calling out specifically which area is being referenced when making different statements.

Abstract:

Line 2: "Others, however, breed poorly, or display stereotypic behaviours indicating stress." Please rephrase as not all stereotypic behavior is indicative of current stress.

Line 6: "Species with large relative brain sizes proved most at risk of oral and whole-body stereotypic behaviour: the first empirical evidence that high intelligence predicts poor captive welfare." Similar, please rephrase as not all stereotypic behavior is indicative of current poor welfare, which is phrased well in the introduction. In addition, statement is generalized, should be specific to Psittaciformes.

Introduction:

Page 2 Line 4: Maybe consider a different word choice other than “dogged”.

Page 2 Line 8-11: These numbers are estimates, this should be more clear for the reader.

Methods:

Page 5 Line 77-79: FDB and stereotypic behaviors should be defined for the reader in the manuscript.

Page 6 Line 107-10: Value that was utilized to determine significance for models should be in the methods.

Results:

Page 8 Line 144: Results are either significant or they are not, results don't “tip into significance.”

Table 2: Trends ($p < 0.01$) should be removed, and only highlight results that are actually significant. The abstract, results and discussion will need to be updated based on this change to the manuscript. Even after removing all non-significant “trends” the results still have at least one significant predictor for each of the outcome variables.

Discussion:

Page 11 Line 223-25: Results don't indicate this idea, this is one possible explanation. This paragraph should be updated to reflect this difference.

Page 12 Line 264: Starting at this point, this will need to be updated, most likely removed after removing all “trends” from the manuscript.

Page 13 Line 285: Intelligence in Psittaciformes, but cannot generalize across all taxonomic groups.

Decision letter (RSPB-2021-1027.R0)

12-Jul-2021

I am writing to inform you that this version of your manuscript RSPB-2021-1027 entitled "Nature calls: Intelligence and natural foraging style predict poor welfare in captive parrots" has, in its current form, been rejected for publication in Proceedings B.

This action has been taken on the advice of referees, who have recommended that substantial revisions are necessary. With this in mind we would be happy to consider a resubmission, provided the comments of the referees are fully addressed. However please note that this is not a provisional acceptance.

Please find below the comments made by the referees, not including confidential reports to the Editor, which I hope you will find useful.

1) A ‘response to referees’ document including details of how you have responded to the comments, and the adjustments you have made.

- 2) A clean copy of the manuscript and one with 'tracked changes' indicating your 'response to referees' comments document.
- 3) Line numbers in your main document.
- 4) Please read our data sharing policies to ensure that you meet our requirements <https://royalsociety.org/journals/authors/author-guidelines/#data>.

Sincerely,
 Professor Gary Carvalho
 mailto: proceedingsb@royalsociety.org

Associate Editor Board Member

Comments to Author:

The reviewers have provided positive feedback and constructive comments that should help to improve the manuscript - particularly in relation to the clarity in use of terminology and the appropriate presentation of the statistical outcomes.

Reviewer(s)' Comments to Author:

Referee: 1

Comments to the Author(s).

A useful and relevant paper that provides some clear applications to industry and the keeping of animals based on ecological knowledge. This is a well-written and well structured paper that has novel and interesting findings. It would benefit from some clarity in the explanations of terms used, the details provided in the statistical modelling and more consideration given to the application of results.

A couple of minor points for consideration:

Take with statements that "this is the first to show something" as there may be evidence somewhere that could come to light in the future.

Avoid the word prove. We support or refute in scientific research.

I can see that you like the term "thrive or fail" but to me, fail does not seem the correct wording for such a statement. Species can fail to adapt to a captive population, but in this sense it is not the individual's failure to cope with its captive husbandry. The husbandry is failing the individual. The animal is not at fault. Please change this word (wherever it occurs in the manuscript).

The abstract provides some really useful explanation of the key results but the concluding statement is quite weak. Could you suggest how they are applied to parrot care?

Also, suggest you remember the word "novel" (findings) because they are currently novel but the paper will age.

The introduction is clear and well explained. Is the word "dogged" correct (line 4)? Something more scientific and less colloquial sounds more appropriate.

The size of the captive parrot population seems huge. How measured? Is this including the budgerigar? And is this an estimation for the entire order?

Species comparison for the grey parrot could be helpful (line 22).

I appreciate that Phylogenetic Comparative Methods are a specialism of this research group but could the authors provide a bit of definition on what they are and what they do? Perhaps something along the lines of the useful diagram (Figure 1) in Mellor et al. 2018 Zoo Biology but using parrots as example species?

Line 30: I read endangeredness as a threat level but this relates to the likelihood of a poor welfare problem being experienced? But then I get to line 61 and see it is related to conservation threat? Please can this be explained more carefully?

If you are discussing how threatened a species is, threat level is probably a better term than "endangeredness".

Line 47: Perhaps explain boredom?

Methods are clear and repeatable and the supplementary information is useful for their understanding.

Line 97: Can you explain the key source for social behaviour?

Line 101: Please explain or define what you mean by encephalisation in this section. Is brain size a proxy for this?

Line 103: Where do captive population sizes come from?

Is aviculture defined as captive populations in managed programmes, i.e. those run by accredited or member organisation zoos? Or for private breeders too?

Line 132: I am not familiar with "leave-one-out" analysis. Can this be explained?

For all models, can you publish the final model that was run?

Line 153: I am struggling to work out why you decide to include and leave out species. What's the biological or ecological decision behind this, and the long term impact on your results?

Line 156: weak effects on...?

Line 157: suggest you explain the correlated predictor here, how strong this was, and what you did.

Figure 1 is useful. How is extensive handling time defined?

Line 189 suggest that you don't lead this paragraph with the ambiguous result. Present the clear result first, that small populations and number of breeding pairs clearly predicted hatch rates. Threat level is unclear. I don't think it is appropriate to try and draw a conclusion from the threat level data given your own explanation of how sensitive it is to specific data points.

Line 160: because the grey parrot is so common in captive populations, is this why it has such an influence? So is the small population of each individual species in this large but varied dataset causing issues with the reliability of the analysis output?

The discussion is thorough and provides a detailed evaluation of results. I would like to see more application of the findings to the actual birds that will benefit from them. Currently this is glossed over or left to the reader to infer.

Line 226-227: I find this sentence hard to follow. And please explain your reasoning for the Type II error.

Line 240-241: Please explain what you mean by better care, and provide some examples of how this can be done (based on your results). Less sunflower seeds, more environmental complexity, for example.

Line 254: Is there a reference for this? Parrots on complete diets can still show abnormal behaviours. How relevant is the comparison to the domestic hen?

Line 272: Perhaps because of genetic drift over time? You compare this finding to the published literature in the crane example stated previously, but I would state your findings and then do the comparison.

Line 291: Why should PCM be used more? Could you add context to the end of your conclusion?

Referee: 2

Comments to the Author(s).

General:

Throughout the manuscript the term captive is utilized when referring to many different situations. This includes pets, aviculture, zoos, etc. and the manuscript could benefit from calling out specifically which area is being referenced when making different statements.

Abstract:

Line 2: "Others, however, breed poorly, or display stereotypic behaviours indicating stress." Please rephrase as not all stereotypic behavior is indicative of current stress.

Line 6: "Species with large relative brain sizes proved most at risk of oral and whole-body stereotypic behaviour: the first empirical evidence that high intelligence predicts poor captive welfare." Similar, please rephrase as not all stereotypic behavior is indicative of current poor welfare, which is phrased well in the introduction. In addition, statement is generalized, should be specific to Psittaciformes.

Introduction:

Page 2 Line 4: Maybe consider a different word choice other than "dogged".

Page 2 Line 8-11: These numbers are estimates, this should be more clear for the reader.

Methods:

Page 5 Line 77-79: FDB and stereotypic behaviors should be defined for the reader in the manuscript.

Page 6 Line 107-10: Value that was utilized to determine significance for models should be in the methods.

Results:

Page 8 Line 144: Results are either significant or they are not, results don't "tip into significance."

Table 2: Trends ($p < 0.01$) should be removed, and only highlight results that are actually significant. The abstract, results and discussion will need to be updated based on this change to the manuscript. Even after removing all non-significant "trends" the results still have at least one significant predictor for each of the outcome variables.

Discussion:

Page 11 Line 223-25: Results don't indicate this idea, this is one possible explanation. This paragraph should be updated to reflect this difference.

Page 12 Line 264: Starting at this point, this will need to be updated, most likely removed after removing all "trends" from the manuscript.

Page 13 Line 285: Intelligence in Psittaciformes, but cannot generalize across all taxonomic groups.

Author's Response to Decision Letter for (RSPB-2021-1027.R0)

See Appendix A.

RSPB-2021-1952.R0

Review form: Reviewer 1

Recommendation

Accept as is

Scientific importance: Is the manuscript an original and important contribution to its field?

Excellent

General interest: Is the paper of sufficient general interest?

Good

Quality of the paper: Is the overall quality of the paper suitable?

Excellent

Is the length of the paper justified?

Yes

Should the paper be seen by a specialist statistical reviewer?

No

Do you have any concerns about statistical analyses in this paper? If so, please specify them explicitly in your report.

No

Do you have any ethical concerns with this paper?

No

Comments to the Author

This is a useful and interesting paper that has been improved and clarified following the first review. The authors have provided useful details to expand on specific areas of the study. All reviewer comments have been answered and suitable extra discussion and evaluation have been provided where relevant. I am happy to approve this paper for publication.

Decision letter (RSPB-2021-1952.R0)

06-Sep-2021

Dear Professor Mason

I am pleased to inform you that your Review manuscript RSPB-2021-1952 entitled "Nature calls: Intelligence and natural foraging style predict poor welfare in captive parrots" has been accepted for publication in Proceedings B.

The referee(s) do not recommend any further changes. Therefore, please proof-read your manuscript carefully and upload your final files for publication. Because the schedule for publication is very tight, it is a condition of publication that you submit the revised version of your manuscript within 7 days. If you do not think you will be able to meet this date please let me know immediately.

To upload your manuscript, log into <http://mc.manuscriptcentral.com/prsb> and enter your Author Centre, where you will find your manuscript title listed under "Manuscripts with Decisions." Under "Actions," click on "Create a Revision." Your manuscript number has been appended to denote a revision.

You will be unable to make your revisions on the originally submitted version of the manuscript. Instead, upload a new version through your Author Centre.

1) A text file of the manuscript (doc, txt, rtf or tex), including the references, tables (including captions) and figure captions. Please remove any tracked changes from the text before submission. PDF files are not an accepted format for the "Main Document".

2) A separate electronic file of each figure (tiff, EPS or print-quality PDF preferred). The format should be produced directly from original creation package, or original software format. Please note that PowerPoint files are not accepted.

3) Electronic supplementary material: this should be contained in a separate file from the main text and the file name should contain the author's name and journal name, e.g. `authorname_procb_ESM_figures.pdf`

All supplementary materials accompanying an accepted article will be treated as in their final form. They will be published alongside the paper on the journal website and posted on the online figshare repository. Files on figshare will be made available approximately one week before the accompanying article so that the supplementary material can be attributed a unique DOI. Please see: <https://royalsociety.org/journals/authors/author-guidelines/>

4) Data-Sharing and data citation

It is a condition of publication that data supporting your paper are made available. Data should be made available either in the electronic supplementary material or through an appropriate repository. Details of how to access data should be included in your paper. Please see <https://royalsociety.org/journals/ethics-policies/data-sharing-mining/> for more details.

<http://datadryad.org/submit?journalID=RSPB&manu=RSPB-2021-1952> which will take you to your unique entry in the Dryad repository.

Once again, thank you for submitting your manuscript to Proceedings B and I look forward to receiving your final version. If you have any questions at all, please do not hesitate to get in touch.

Sincerely,
Professor Gary Carvalho

Reviewer(s)' Comments to Author:

Referee: 1

Comments to the Author(s).

This is a useful and interesting paper that has been improved and clarified following the first review. The authors have provided useful details to expand on specific areas of the study. All reviewer comments have been answered and suitable extra discussion and evaluation have been provided where relevant. I am happy to approve this paper for publication.

Sincerely,

Proceedings B

Decision letter (RSPB-2021-1952.R1)

08-Sep-2021

Dear Professor Mason

I am pleased to inform you that your manuscript entitled "Nature calls: Intelligence and natural foraging style predict poor welfare in captive parrots" has been accepted for publication in Proceedings B.

Data Accessibility section

Open Access

Paper charges

Sincerely,
Editor, Proceedings B
mailto: proceedingsb@royalsociety.org

Appendix A

Responses to referees

Note that all line numbers refer to the non-tracked version

Ref 1

Comments to the Author(s).

A useful and relevant paper that provides some clear applications to industry and the keeping of animals based on ecological knowledge. This is a well-written and well structured paper that has novel and interesting findings.

Many thanks for your nice comments here (and the constructive ones throughout).

It would benefit from some clarity in the explanations of terms used, the details provided in the statistical modelling and more consideration given to the application of results.

Your suggestions were very helpful, and we hope we've complied with them as you envisaged.

A couple of minor points for consideration:

Take with statements that "this is the first to show something" as there may be evidence somewhere that could come to light in the future.

Very true -- though we are certain that to date, there has been no empirical evidence that intelligence can put species at risk of poor welfare.

Avoid the word prove. We support or refute in scientific research.

Totally agree; in the abstract we were using it in the non-scientific sense of "to demonstrate" or "to show" (to save words), but can now see that this is needlessly confusing. Text is now changed accordingly to "*were most prone to*".

I can see that you like the term "thrive or fail" but to me, fail does not seem the correct wording for such a statement. Species can fail to adapt to a captive population, but in this sense it is not the individual's failure to cope with its captive husbandry. The husbandry is failing the individual. The animal is not at fault. Please change this word (wherever it occurs in the manuscript).

Totally agree with your sentiment here. If we accidentally implied that animals are at fault, that is bad: thank you for pointing this out. In the abstract, we removed that clause (saving words: nice as the limit is 200). And in line 1 of the Introduction, we replaced “fail” with “struggle”, as we also did in the very last line of the paper.

The abstract provides some really useful explanation of the key results but the concluding statement is quite weak. Could you suggest how they are applied to parrot care?

We reordered it so that the penultimate sentence emphasises parrot husbandry more (*“Together, results suggest that more naturalistic diets would improve welfare and intelligent psittacines need increased cognitive stimulation.”*), but we are very constrained by the word limit (200 words!). We also do want to highlight the future value of PCMs (the last line), as the application of this ecological/evolutionary method to welfare issues is still very new and with great untapped potential.

Also, suggest you remember the word "novel" (findings) because they are currently novel but the paper will age.

Good point; removed

The introduction is clear and well explained.

Thankyou!

Is the word "dogged" correct (line 4)? Something more scientific and less colloquial sounds more appropriate.

It's a bit poetic, you're right. Changed to “prone to” (new line 6).

The size of the captive parrot population seems huge. How measured? Is this including the budgerigar? And is this an estimation for the entire order?

It is estimated by Jamie Gilardi, director of the World Parrot Trust. We believe it does include budgerigars.

It sounds large, but yes it's for the entire order, spanning c. 350 species. So on average the number of individuals per species is only c. 285,000. (And for context, this total for all 350 species is a fraction of even just the no. of farmed turkeys in the world – around 500 million;

and of course of the global human population – 7800 million. So quite small compared with these).

Species comparison for the grey parrot could be helpful (line 22).

Good idea; added Senegal parrots as a counter-example (at new line 25).

I appreciate that Phylogenetic Comparative Methods are a specialism of this research group but could the authors provide a bit of definition on what they are and what they do? Perhaps something along the lines of the useful diagram (Figure 1) in Mellor et al. 2018 Zoo Biology but using parrots as example species?

We are hitting the page limit for this journal so are unable to add too many details, but we augmented the text to add (the underlining indicating new text): *“Phylogenetic comparative methods (PCMs) developed by evolutionary biologists can reveal why species show such variation. These allow multi-species datasets to be statistically analysed to identify attributes that predispose certain taxa to problems in captivity, while controlling for similarities arising from evolutionary relatedness”* (New lines 27-30). We also added your suggested figure to the Supported Materials (new Fig S1), with some explanatory text: nice idea so thank you.

At lines 302-303, we also say *“PCMs – with their abilities to interrogate multi-species datasets and address hypotheses that would otherwise be challenging to test”*.

Line 30: I read endangeredness as a threat level but this relates to the likelihood of a poor welfare problem being experienced? But then I get to line 61 and see it is related to conservation threat? Please can this be explained more carefully? If you are discussing how threatened a species is, threat level is probably a better term than "endangeredness".

Threat level is one of the variables we use to test this hypothesis, so we want to stick to "endangeredness" for the broader concept. But we do see it could be worded more clearly, so thanks for pointing this out. Lines 35-36 now say *“A third is a species’ conservation status (endangeredness). This can predict vulnerability to both poor welfare [2, 32] and problems induced by small population sizes per se (e.g., inbreeding [35]).”*

And lines 69-73 now say *“The final type of potential risk factor is an endangered conservation status. Traits common in threatened species – especially timidity, wide-ranging lifestyles and low behavioural plasticity – may put them in “double jeopardy”: both vulnerable in the wild and at risk of stress when captive [2, 32]. Furthermore, the small population sizes of rare species can cause further issues”*.

Lines 109-110 also now say: ***“Our two measures of endangeredness were threat level according to IUCN Red List categories, and ex situ rarity”***

Line 47: Perhaps explain boredom?

Ormrod and Morris (who we cite here), were using this term rather colloquially so we left as is here (plus we don't want to make this long section longer). But we now define the term at new lines 250-251, where a precise meaning is important: ***“promoting boredom (aversive states caused by monotony [38])”***

Methods are clear and repeatable and the supplementary information is useful for their understanding.

Thankyou! We've also added a bit more detail to the Supplement to make it more useful still (new Fig S1; plus in Section 6, details of how to access species-typical data from two online datasets).

Line 97: Can you explain the key source for social behaviour?

As cited in Table 1, they were:

[70] Beauchamp, G. & Fernandez-Juricic, E. 2004 Is there a relationship between forebrain size and group size in birds? *Evolutionary Ecology Research* 6, 833-842

and
[71] Munshi-South, J., Wilkinson, G. S. & Vega Rivera, J. H. 2006 Diet influences life span in parrots (Psittaciformes). *The Auk* 123, 108-118. (DOI:10.1642/0004-8038(2006)123[0108:DILSIP]2.0.CO;2).

Both [70] and [71] contained data on feeding groups, and [71] also had data on roosting group sizes.

Line 101: Please explain or define what you mean by encephalisation in this section. Is brain size a proxy for this?

Relative brain size (i.e. brain size corrected for allometric effects of body size). This is now defined on new line 54 (and was also in Table 1, and is still there).

Line 103: Where do captive population sizes come from?

They come from the aviculture dataset (see Table 2); thus from the private breeders who hold the vast majority of breeding parrots.

Is aviculture defined as captive populations in managed programmes, i.e. those run by accredited or member organisation zoos? Or for private breeders too?

No, aviculture means breeding centres, which are private. Zoo populations are small, and hardly any parrots in zoos are managed using studbooks (under a dozen species).

For clarity on this, at lines 76, 101, and 111 we have now changed “aviculture” to “private breeding centres”.

Line 132: I am not familiar with "leave-one-out" analysis. Can this be explained?

New lines 142-147 now say (new bits being underlined): “To assess whether any significant findings merely reflected outliers, we used a custom version of the `influ_phylm` function within the ‘sensiPhy’ package [102]. This performs “leave-one-out” deletion analyses (removing each species in turn and recalculating the intercept, slope and corresponding *p* value for each parameter), a species being deemed ‘influential’ if its removal yielded a standardised difference >2. This revealed whether any results critically relied on the influence of just one or two particular datapoints”.

For all models, can you publish the final model that was run?

Throughout the manuscript and supplementary material, we have followed the principle that anyone should be able to replicate our methods (including all the decisions that led to the final models). Thus our methods are very transparent and replicable, and the final models and their rationales are all summarised in Table S6. Furthermore, the R scripts are provided in the relevant Dryad repository (see Section 6 of the Supporting Material), where the full dataset itself is also freely available to anyone wanting to check our analyses. (Note that it’s possible Dryad may not make this link active until the paper is in press, however)

Line 153: I am struggling to work out why you decide to include and leave out species. What's the biological or ecological decision behind this, and the long term impact on your results?

Apologies, this was rather buried in the Supporting Material. A species was deemed 'influential' if its removal yielded a standardised difference >2: an arbitrary but objective rule implemented by the 'Sensiphy' package (and this is now added to the main text, see below). These are thus essentially species that have a high leverage effect on the results. If significant effects vanished when such species were removed, the significant effect was deemed non-robust / unreliable (because critically dependent on what was effectively an outlying datapoint). Table S7 now includes a fuller description of our conclusions too: we hope this is helpful.

Line 156: weak effects on...?

Removed, now that trends are no longer discussed (following Referee 2's advice)

Line 157: suggest you explain the correlated predictor here, how strong this was, and what you did.

Again, removed, now that trends are no longer discussed (following Referee 2's advice)

Figure 1 is useful. How is extensive handling time defined?

In the absence of time or energy budget data (there's just too little detailed fieldwork for most species), we inferred it from typical diets. This was done blind to captive behaviour (to avoid the potential for bias), in consultation with parrot experts (especially Cathy Toft), and following another author's published methods (Mettke-Hoffman). Table 1 thus says:

"Time budget data were unavailable. Therefore, time spent in the wild on food-search and/or -handling [cf. 72] was inferred from diet-types reported in the EltonTraits avian foraging database [73] and other sources (75-77):

% natural diet requiring extensive locomotor/visual food search: reliance on food items that are naturally patchily distributed in space and time, and/or scarce or inconspicuous (based

on [74-76] and adapted here in line with EltonTraits' dietary categories), namely invertebrates, nectar and pollen, fruit, and tree seeds/nuts.

% natural diet requiring extensive food handling: reliance on food items requiring extensive oral manipulation (based on [74-76] and adapted here), namely invertebrates and tree seeds/nuts."

Lines 259-260 also always said: *"reliance on food items in the wild that need extensive handling (inferred from diet type, since cross-species data on handling time budgets were unavailable)"*

Line 189 suggest that you don't lead this paragraph with the ambiguous result. Present the clear result first, that small populations and number of breeding pairs clearly predicted hatch rates. Threat level is unclear. I don't think it is appropriate to try and draw a conclusion from the threat level data given your own explanation of how sensitive it is to specific data points.

Agreed; this section has been re-ordered and shortened accordingly.

Line 160: because the grey parrot is so common in captive populations, is this why it has such an influence? So is the small population of each individual species in this large but varied dataset causing issues with the reliability of the analysis output?

No, there is no weighting for population size: each species is treated as a single datapoint in all analyses. (But note that this sentence has also since been deleted, as trends are no longer discussed).

The discussion is thorough and provides a detailed evaluation of results.

Thanks so much.

I would like to see more application of the findings to the actual birds that will benefit from them. Currently this is glossed over or left to the reader to infer.

This is inevitable because PCM results can generate new broad principles (their strength), but not precise practical advice tailored for each and every species. They thus show for the first time that being "brainy" is a risk factor, suggesting that something to do with learning and cognition is important for captive welfare... but *precisely* what this is, they cannot reveal. We hope new empirical work, inspired by these findings, will now figure this out. The results also

show that something about natural diets that require much handling is also important for FDB, but whether this is because birds need enrichments that require manipulation/chewing, or whether this reflects some other properties of these diets (e.g. how they affect the gut microbiome), is again impossible to say. But again we hope our results inspire new empirical studies.

In addition, the abstract is now more focussed on captive care, and now says: *Together, results suggest that more naturalistic diets would improve welfare, and intelligent psittiforms need increased cognitive stimulation*".

In the Discussion line 372 also now says *"captive diets may lack the nutrients or cellulose/chitin present in nuts, tree seeds and invertebrates"*, to be a bit more specific.

Note that the data we collated also provide a freely accessible resource for anyone wanting to look up the characteristics of any species they're concerned about. We have added a note to this effect in the Results (new lines 112-113), and added a note at the end of Section 6 of the Supplementary Material too, to help make these data more accessible.

Line 226-227: I find this sentence hard to follow. And please explain your reasoning for the Type II error.

Rewritten; we hope it is now clearer with the greater detail added. These sections therefore now read as follows, at lines 226-234 (underlining indicating new text):

"... but instead in how they display it, with smaller-brained species expressing poor welfare hypo-actively (e.g. with apathy) instead of hyper-actively via stereotypic behaviour" and "Alternatively, however, that relatively large brains did not predict poor reproduction could well be a false negative result. This seems likely because, while reproductive problems can reflect stress [20, 112], they have poor specificity as stress indicators, being affected by multiple other factors (including genetic bottlenecks [cf. 17, 63], a lack of appropriate imprinting opportunities, and artificial incubation [e.g., 125]): all factors that would add noise to our analyses which we could not parse out (so increasing risks of Type II error)."

Line 240-241: Please explain what you mean by better care, and provide some examples of how this can be done (based on your results). Less sunflower seeds, more environmental complexity, for example.

We have now said at new lines 247-256 (the underlined bits being new):

“being given care better tailored for intelligent species. As for precisely what large-brained species lack in captivity, opportunities for exploration, learning and agency are all possibilities. Their absence could cause the diverse array of SBs shown by large-brained parrots by compromising normal brain development (cf. [21]); by promoting states of boredom (aversive states caused by monotony [38]) and resulting attempts to self-stimulate [108,111]; and/or by enhancing birds’ motivations to escape (as underlie some mammalian SBs [21]; a concerning possibility given the invasiveness of these larger-brained parrots). Opportunities for cognitive stimulation must therefore urgently be investigated, to identify the most effective ways to improve these parrots’ well-being [e.g., 48, 116].”

We hope this better conveys the ways in which being intelligent may lead to welfare problems, but also we hope it now better conveys that identifying the best solutions for these birds now needs some practical empirical research.

Line 254: Is there a reference for this?

We are referring to our own results here.

Parrots on complete diets can still show abnormal behaviours.

But do they still show FDB? And if they still have FDB, it is at least reduced by feeding complete diets? This is the type of empirical info. we don’t presently have, and very much need.

(Also, SBs can become harder to treat with time/age: another reason why principles that allow them to be pre-empted [rather than just reacted to] are important).

How relevant is the comparison to the domestic hen?

Very relevant! Self- and allo-plucking behaviours in mammals seem derived from social grooming (e.g. they are predicted by natural group size in Primates; see ref [39]). But feather-plucking in hens is derived from foraging, which makes this newly revealed avian convergence across Galliformes and Psittaformes pretty intriguing. Furthermore, it is relatively easy to conduct invasive, controlled experiments with good Ns using chickens (unlike for parrots), and such research has already showed that feather ingestion rectifies slow gut motility, in part due to effects on the gut microbiome. This cool agricultural work could perhaps lead to novel remedies for feather-plucking in hens (e.g. aimed at altering gut microflora), which perhaps may ultimately prove useful for parrots too. So please let us leave these references in the paper: captive birds are captive birds after all!

(Work on poultry can also help us understand links between relative brain size and adaptation to captivity too; see reference [109] and new lines 236-237).

Line 272: Perhaps because of genetic drift over time? You compare this finding to the published literature in the crane example stated previously, but I would state your findings and then do the comparison.

With respect, after a lot of thought, we'd like to leave as is, because the broad "endangeredness" idea (which came from Derrickson and Snyder) suggested two separate hypotheses – that species-typical traits conferring vulnerability in the wild also confer vulnerability to poor welfare in captivity (something we did not find support for), and also that small populations are at risk of breeding problems for genetic reasons (something we did find support for).

Line 291: Why should PCM be used more? Could you add context to the end of your conclusion?

Good idea. While also trying not to be too wordy, we added "*.. – with their abilities to interrogate multi-species datasets and address hypotheses that would otherwise be challenging to test --...*".

A few lines up (lines 297-298) we also suggested new research ideas that are "*topics ideal for future PCMs*", by way of illustration.

Ref 2

Comments to the Author(s).

General:

Throughout the manuscript the term captive is utilized when referring to many different situations. This includes pets, aviculture, zoos, etc. and the manuscript could benefit from calling out specifically which area is being referenced when making different statements.

Thank you for your thoughtful comments; they really helped improve the paper.

We use the term “captive” to refer to all these conditions because all of them constrain flight, migration, social contact, parental care, mate choice, and diet choice (etc.) (and also provide veterinary care and protection from predators). With respect, we’d like to keep this term because it’s accurate (they are all literally captive situations); this language is consistent with other published papers (it’s thus not unorthodox); and using this general word reduces wordiness too (otherwise the text risks becoming long and unwieldy).

Of course there are difference in degrees of constraint, both within and across different types of captivity: zoos across the world vary greatly in the quality of their care, as do private breeders and pet owners; but we worked hard to take this variation into account, where we had data, to ensure this did not generate confounds (see Tables S3 and S4). And to line 89-90 we therefore added “*species-typical demographic and husbandry characteristics (see below and Table S1), which could potentially be highly influential and even act as confounds”.* In one example where the species differences could have reflected housing differences, we added the clarification “*Others in contrast, despite living in similar conditions, are...” (at line 15).*

We should add that this term is also not meant to be pejorative; we do say “*Many wild animal species enjoy impressive lifespans and breeding success when kept in zoos, breeding centres or people’s homes [1]*”, and we refer to some captive species as “thriving”.

Abstract:

Line 2: “Others, however, breed poorly, or display stereotypic behaviours indicating stress.” Please rephrase as not all stereotypic behavior is indicative of current stress.

We never meant to say that stereotypic behaviour indicates *current* stress; however it does indicate poor welfare at some point in an animal’s life (or even cumulatively over the lifetime, suggests evidence from research deer mice, elephants in zoos and laboratory

primates), with early weaning, repeated aversive experiences over the lifespan, and current frustration all being key risk factors.

The abstract has been edited to refer to “*behavioural and reproductive problems*” (consistent with Tables 1 and 2). Elsewhere, however, we do retain the word ‘stress’ because this encompasses developmental stress as well as current stress.

Line 6: “Species with large relative brain sizes proved most at risk of oral and whole-body stereotypic behaviour: the first empirical evidence that high intelligence predicts poor captive welfare.” Similar, please rephrase as not all stereotypic behavior is indicative of current poor welfare, which is phrased well in the introduction. In addition, statement is generalized, should be specific to Psittaciformes.

Now rephrased to “*Parrot species with relatively large brains were most prone to oral and whole-body stereotypic behaviour*”.

However, the reference to poor welfare is deliberately left in, since stereotypic behaviours most definitely indicate sub-optimal husbandry (even if sometimes they are ‘scars of the past’ rather than evidence of current negative affect): see much evidence for this (and our summary at lines 221-223, which highlights their good specificity if poor sensitivity), including the selection of texts we had room to cite:

[21] Mason, G. 2006 Stereotypic behaviour in captive animals: fundamentals and implications for welfare and beyond. In *Stereotypic Animal Behaviour: Fundamentals and Applications to Welfare* (eds. G. Mason & J. Rushen), pp. 325-367, Second ed. Wallingford, CAB International.

[22] Grandin, T. & Deesing, M. J. 2014 Genetics and animal welfare. In *In Genetics and the Behavior of Domestic Animals* (pp. 435-472, Second ed.

[23] Miller, L. J., Ivy, J. A., Vicino, G. A. & Schork, I. G. 2018 Impacts of natural history and exhibit factors on carnivore welfare. *Journal of Applied Animal Welfare Science*, 1-9. (DOI:10.1080/10888705.2018.1455582).

[24] Mellor, D. J., Hunt, S. & Gusset, M. 2015 *Caring for Wildlife: the World Zoo and Aquarium Animal Welfare Strategy*. (p. 87. Gland, Switzerland.

[106] Latham, N. R. & Mason, G. J. 2008 Maternal deprivation and the development of stereotypic behaviour. *Applied Animal Behaviour Science* 110, 84-108. (DOI:http://dx.doi.org/10.1016/j.applanim.2007.03.026).

[107] Greco, B. J., Meehan, C. L., Hogan, J. N., Leighty, K. A., Mellen, J., Mason, G. J. & Mench, J. A. 2016 The days and nights of zoo elephants: using epidemiology to better understand stereotypic behavior of African elephants (*Loxodonta africana*) and Asian elephants (*Elephas maximus*) in North American zoos. *PLoS One* 11, e0144276.

[108] Swaisgood, R. & Shepherson, D. 2006 Environmental enrichment as a strategy for mitigating stereotypies in zoo animals: a literature review and meta-analysis. In *Stereotypic Animal Behaviour: Fundamentals and Applications to Welfare* (eds. G. Mason & J. Rushen), pp. 256-285. Wallingford, UK, CAB International.

Introduction:

Page 2 Line 4: Maybe consider a different word choice other than “dogged”.

Agreed; thank you for spotting this odd choice of verb! Now changed to “prone to”; thanks for that.

Page 2 Line 8-11: These numbers are estimates, this should be more clear for the reader.

This is now lines 26-32, and respectfully we think it is clear that these are estimates because they are just expressed in terms of orders of magnitudes (“thousands”, “tens of thousands”, “tens of millions”). Also, these values do come from surveys (they are empirical, not just guestimates).

Methods:

Page 5 Line 77-79: FDB and stereotypic behaviors should be defined for the reader in the manuscript.

FDB was defined at line 22, earlier in the paper; and SB is now defined at new line 100 (thanks for spotting that omission).

Page 6 Line 107-10: Value that was utilized to determine significance for models should be in the methods.

Good suggestion; now added to line 118.

Results:

Page 8 Line 144: Results are either significant or they are not, results don’t “tip into significance.”

Now substantially edited, so this has been removed.

Table 2: Trends ($p < 0.01$) should be removed, and only highlight results that are actually significant. The abstract, results and discussion will need to be updated based on this change to the manuscript. Even after removing all non-significant “trends” the results still have at least one significant predictor for each of the outcome variables.

Done throughout (including in all the supporting tables of the online supplement).

Also much more detail has now been added to Table S7 (on the “leave-one-out” tests), and results here support your advice: all former trends turned out to be non-robust, i.e. reliant on just one-two species with disproportionate leverage). Very helpful idea!

As a result the key messages are much stronger now (and we hope the Results are easier to read as well), so thank you for this really useful suggestion.

Discussion:

Page 11 Line 223-25: Results don't indicate this idea, this is one possible explanation. This paragraph should be updated to reflect this difference.

Yes, perhaps not the best verb: “indicate” has now been changed to “reflect”. This is a complicated section to write succinctly because there is good evidence to argue that species with high SB have poor welfare, but whether species with minimal/no SB therefore automatically have good welfare is less clear (because they may have great welfare, or they may be displaying equally concerning hypo-active responses instead, as an alternative response style). But we hope it is clearer now.

Page 12 Line 264: Starting at this point, this will need to be updated, most likely removed after removing all “trends” from the manuscript.

Done – the Discussion has been edited accordingly.

Page 13 Line 285: Intelligence in Psittaciformes, but cannot generalize across all taxonomic groups.

Yes, fully agreed, and we do make that clear. For example, we suggest that more work is now needed on other taxa, like Cetaceans and Primates, to see if the same effects apply to these groups too (since this should not be assumed). See new lines 297-298.